# Sustainable Environmental Development from the Regional Perspective—The Interesting Case of Poland

Eva-Luz Tejada-Gutiérrez [1], Zofia Koloszko-Chomentowska [2], Mariantonietta Fiore [3] and Alessia Spada [3,*]

1 Department Ciències Mèdiques Bàsiques, Universitat de Lleida, 25003 Lleida, Spain
2 Department of Management Economy and Finance, Bialystok University of Technology, 15-351 Bialystok, Poland
3 Department of Economics, University of Foggia, 71121 Foggia, Italy
* Correspondence: alessia.spada@unifg.it

**Abstract:** The EU regional development policy aims to reduce through its cohesion policy the socio-economic development disparities between its most developed regions and peripheral regions. Research shows that despite this strategy, the economic development of EU countries in the spatial system is not homogeneous. In addition, contemporary strategies take into account economic development with respect to environmental protection. The goal of this paper was to assess sustainable regional development in Poland, taking into account environmental and innovative activity indicators, as well as mobility and accessibility in remote areas. To recognize the effects of supporting sustainable development, the evaluation was carried out considering the indicators from the Statistics Poland platform, concerning 2011 and 2021, and related them to economic development, the environment, mobility and accessibility of 16 regions. Comparing results obtained with the K-means clustering method with those obtained with the partitioning around medoids method, it was possible to verify, with greater reliability, the migration of regions into different clusters. Results showed that development of all the regions was significant yet highly disproportionate, thus highlighting the consolidated presence of dynamically growing regions versus peripheries, which means further regional polarization. An educational cross-sectorial approach can play a crucial role in promoting green pro-innovative development.

**Keywords:** sustainable-economic-social development; Poland; partitioning around medoids (PAM) method; K-means clustering; center-periphery polarization

## 1. Introduction

In recent years, many European countries have investigated the rapid growth of large cities, which are the center of socioeconomic life, and the much slower development of rural areas, particularly of those at greater distances from urban centers. This phenomenon is called center-periphery polarization [1–3]. The persistent nature of inter-regional differentiation leads to the separation of central and peripheral regions. Persisting differences in the economic potential of individual regions are one of the fundamental problems of the modern economy. The European Commission refers to 'problem regions' as the less developed regions [4]. Despite being interesting areas, they are characterized by a lower level of economic potential, in comparison to more developed regions. The causes of unequal development vary widely. They include, among other things, historically shaped neglect of economic development, demographic structure, and locations near economic centers. The persistent nature of developmental disproportions leads to consolidation of the divisions in the country's spatial structure. The size and geographic location of regions have strategic significance for the national spatial development policy [5]. Dynamic globalization processes that arose from the use of information technologies have contributed to making the problem of local development widespread [6]. Territorial development is

always limited by the lack of capital, the dynamics of its utilization and the localization of its distribution within a given territory [7].

The economic development of every country should be sustainable, and characterized by long-term invariability of parameters. On the macro scale, it should contribute to the creation of a strong economy in the given country. On the micro scale, it entails improving the quality of life of households and the generation of profits for economic entities. Various concepts that describe processes of initiating and stimulating regional development have been created on this foundation. Localization theories are an important group among the many theories that have been formulated on this topic. The role of modern technologies in the search for optimal localization conditions is also currently being indicated [8].

In modern theories of regional development, besides classical growth factors, the role of innovations and the level of innovation arising from their implementation have been highlighted [2,4]. As stated by the Organization for Economic Co-operation and Development (OECD), regions are trying to promote their sustainable economic development increasingly by means of supporting innovation. Innovation can be considered a spin of development, and a crucial means for coping with global challenges. In this contest, regions are seeking to promote their economic development through supporting innovation. Therefore, stable and long-term improvements in the competitiveness of entities or regions are linked to their innovative activity and capacity. This has particular significance in the case of peripheral, poorly developed regions.

Paradoxically, regions with a relatively low capacity for absorbing financing in comparison to more advanced regions show delayed investment, but exhibit greater demand for innovations [9]. The innovations in demand are technological, social and institutional in nature [10,11]. Still, a special role is assigned to new technologies in solving problems related to the mobility and integrity of regions [12]. In fact, innovative concepts of mobility are being developed, along with the development of technology, which lead to improvements in the living conditions of residents in rural areas. From a social perspective, mobility plays a key role in mitigating the process of human migration from rural areas, and of social exclusion [13,14]. Hence, this is an important part of integrating local communities [15–18]. With the scope of preventing exclusion and improving mobility, various forms of financial support are proposed [19]. People with low incomes are interested in financial incentives, above all [20].

In this context, the goal of this paper was to assess the sustainable regional development of EU regions, taking into account environmental and innovative activity indicators, mobility and accessibility of remote areas of Poland. This choice of indicators is widely justified in the context of modern challenges related to sustainable development.

Therefore, the paper broadens the current state of knowledge on the subject of regional development in so-called peripheral regions, and tries to fill in gaps in the literature; in addition to economic aspects, it also accounts for the environmental aspects (with a social perspective, too) more broadly, which is not an approach frequently encountered in recent literature.

The research was developed following this structure: the next section focuses on the background regarding the problem of unbalanced development of regions within the context of new sustainable challenges. Then, the paper explains and defines materials and methods implemented to reach the aims of the study. Results section describes what the research found, and how the results can provide new insights. Finally, sections dedicated to conclusions and future research close the paper, highlighting the possible implications of supporting transformation processes in regional development.

*Literature Review*

The problem of regionally unbalanced development is a significant obstacle to achieving sustainable growth of the entire national economy. This issue is particularly acute in the modern realities of environmental and technological trends. The differences in the innovation and investment attractiveness of regions depend on the specifics of each

territory, which are formed from geographical, institutional, economic, technological, and other influences [21]. Considering these factors, and searching for common patterns of the socio-ecological environment of territorial formation, is of great scientific and practical importance for building a balanced territorial system in the country.

One of the more important economic and environmental challenges of the modern world is sustainable development, and the transition to a low-emissions economy. This is convergent with the development goals of EU member states, and is set forth in the European Green Deal [8,11]. The key priorities of the European Green Deal are the following: protection of natural capital; transition to a resource-efficient economy; and protection of humans against environmental pressures. The benefits of a low-emissions economy pertain to all societies, regardless of their level of affluence [22]. Climate change is the biggest challenge of our times, and it presents the opportunity to build a new economic model [12,23].

Sustainable development requires compromises between economic, environmental and social objectives. The creation of such development is considered to be a function of social well-being, which integrates fundamental economic needs (economic capital), and the integrity of ecosystems (environmental capital) and social systems (social capital) [22]. Growing interest in social aspects is being observed in the debate on sustainable development. Social capital is a multi-functional factor, and serves as the basis for collective action, which is why it should be taken into account when assessing social well-being, in addition to environmental quality and income per capita [24–26].

Local and regional development occurs through a combination of economic development and respect for the environment [27]. Therefore, the goals of sustainable development concern improvements in people's living conditions, and protection of the environment against degradation, in order to support the needs of current and future generations [23,25].

In the Strategy for Responsible Development, it was assumed that the individual potentials of regions would be utilized. One example is Eastern Poland, where developmental processes occur at a slower pace. Actions related to improvements in the country's attractiveness have been planned. These actions do not carry environmental threats, so long as this region's most important asset is not lost, namely, its relatively high environmental quality. For example, measures involving the creation of large-sized farms raise certain doubts, as they may have a negative impact on biodiversity. It was assumed that, by 2030, the focus of regional policy on problem areas will be increased. This pertains to the creation of conditions toward increase the income of the population, while simultaneously enhancing economic, social and environmental cohesion. Failure to perform the accepted tasks could result in further marginalization, and the depopulation of certain regions, which is why projects covered by the strategy are strategic tasks for the government.

Therefore, the paper aimed to investigate the actual context of regional development processes, in order to recognize the effects of supporting sustainable development.

In past decades, concepts relating to Keynes's doctrine have been appearing to a greater extent. These concepts refer to a broader range of factors that influence the development of regions, and call for interventionism on the part of public authorities, which is due to the accumulation of differences between regions. The EU's regional development policy strives to reduce differences in the level of socio-economic development between the most developed regions in the Community, and its peripheral regions. This is implemented through its cohesion policy [4]. Both debates and specific actions concerning sustainable development and equalization of differences between regions have been undertaken for many years. According to the inter-regional convergence theory, differences between developed and peripheral regions should disappear over time. Research [1,3,4,28] indicates that, despite undertaken actions, socio-economic development in the spatial system has varied, and the differences are even deepening. Disproportionate levels of development are linked to the intensive development of cities and their suburbs, while rural areas are depopulating [29]. Peripheralization, perceived as the growth of social and spatial inequalities, is a process that is still progressing [1].

The literature most frequently analyzes the process of spatial differentiation of development through the lens of economic potential. Less attention is dedicated to socio-environmental aspects of regional and local development [2].

Currently, innovative activity is a fundamental stimulus for development in all of these aspects, and improvements in mobility, particularly of areas distant from socio-economic centers, prevent social exclusion. Indeed, improvements in the mobility of regions is one of the most important factors in their development [30–32]. The bicycle is the most sustainable form of movement, and should serve as the basis for the majority of mobility. However, they are of secondary significance as a means of transportation [33,34]. In this context, social expectations concerning mobility in Poland concern innovative solutions, particularly with regard to public transportation, but clearly not only in regards to it [35,36]. This can be considered an answer to technological progress, and on the other hand, to changing demographic trends and social behaviors in Poland.

These innovative activities also concern environmental protection and water management [36]. According to the EC report, the weakest aspect is the lack of investment in environmental technologies [37].

Spatial disproportions give rise to fears concerning further regional polarization. At the same time, the sustainable development concept is not conflict-free, and manifests the importance of implementing solutions that involve a cross-sectorial approach [38–40]. There are similar fears concerning all of Eastern Europe [2]. Education and strengthening human capital should play a special role in promoting more poorly developed regions as well as pro-innovative development [41].

Current global processes, such as population growth, urbanization, and globalization, have resulted in natural resource depletion issues and environmental issues on a regional level [42]. Adamashvili and colleagues [43] argue that "trust-based collaborations promoted by regulations and local development plans (LDPs)" play a crucial role in overcoming this problem. Some researchers [44,45] focused on the importance of discussing perspectives for entrepreneurship, growth and development separately on a regional level. According to some authors [46,47], this will help regions reach sustainable development goals, international cooperation and minimize gaps between the development of different regions.

In line with this challenging stimulus, the REInA (Rural European Innovation Area) was born. It is a pan-European open platform aimed at collecting innovative initiatives to endorse a new Rural European Innovation Area, by pushing development of rural areas by means of a fair, green and sustainable management.

The actions of the Polish government are focused on development of the most poorly developed regions, under the assumption that more rapid growth of macro-regions will contribute to territorially sustainable development of all of Poland. Some actions are already being implemented through a special European fund known as "Polska Wschodnia" [Eastern Poland]. One of the more important tasks is sustainable mobility, as low transportation accessibility is recognized as the main barrier to the development of this part of the country. Projects concern redevelopment, modernization of railway lines and roads, as well as improvements in their technical parameters. The choice of environmentally friendly means of mobility and investment outlays for environmental protection indicate the orientation of a development strategy that is consistent with respect for the environment [39]. Poland has benefited from many programs supporting regional development, mainly financed with EU funds, which is why it seems interesting to learn about the effects of this support. EUR 690 million has been allocated for this goal, and works are already at an advanced stage. It is expected that these investments will have a positive influence on revitalizing entrepreneurship, and improving access to the labor market and public services. It will be possible to assess the effects of the measures over a longer time horizon, and this is the subject of further research.

## 2. Materials and Methods

The research made use of mass statistical data from the Local Data Bank (LDB) [48], the Poland's largest database of the economy, society and the environment, with data from 1995. The LDB delivers over 40 thousand statistically grouped thematic indicators according to the NUTS nomenclature [49]: macro-region, region and sub-region. Figure 1 summarizes the flow of the methodology.

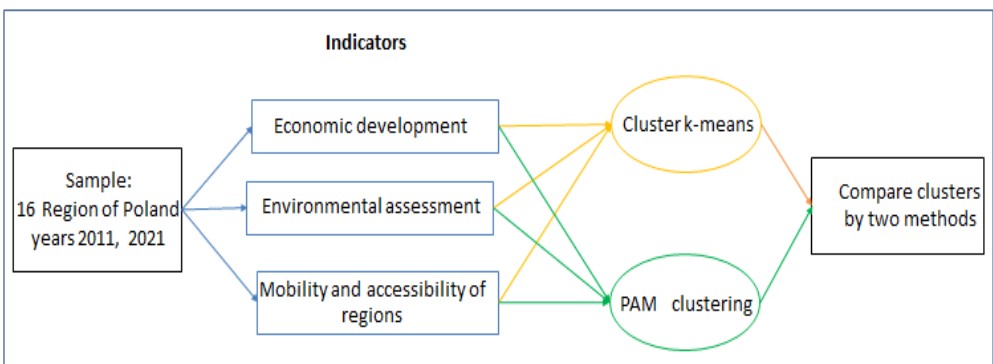

**Figure 1.** Methodology flowchart.

The assessment of the level of regional development takes into account the indicators characterizing various aspects of Poland's development: economic, social and environmental. The assessment was made based on the following indicators:

X1—number of economic entities per 1000 inhabitants

X2—gross salary (Poland = 100)

X3—value of EU funds per inhabitant (PLN)

X4—fixed asset outlays for environment protection per inhabitant (PLN)

X5—fixed asset outlays for water protection per inhabitant (PLN)

X6—innovation expenditure in relation to gross fixed assets (%)

X7—expressways and highways at 1000 km$^2$ (km)

X8—railway lines per 100 km$^2$ (km)

X9—cycle paths per 100 km$^2$ (km)

Indicators X1, X2 and X3 pertain to the level of economic development. This group also includes the ratio of outlays for innovations to gross fixed assets (X6). Innovation activity is indicated as the primary factor affecting the progress and development of regions. The following indicators were applied for environmental assessment: X4 and X5. The selected indicators are a measure of actions taken to protect the natural environment. Accounting for environmental indicators in the assessment of regional development seems to be justified. Creating development consistently with principles of respect for the environment is an essential condition for implementing a sustainable development strategy.

The analysis also took into account indicators characterizing the mobility and accessibility of regions (X7, X8 and X9). Empirical evidence consistently indicates that transportation infrastructure has a substantial influence on the economic development of regions [17,49]. Indicators, such as railway lines (X8) and cycle paths (X9), are proof of sustainable mobility.

The analysis takes into consideration all regions of Poland, which are 16 in number. Regions differ in their levels of development, and 5 of them are considered to be peripheral regions.

The data concern two research periods, 2011 and 2021, which made it possible to learn about changes in the level of socio-environmental development, and to assess the effects of support for regional development. The adopted time horizon allowed for such an assessment.

In order to categorize Poland's regions, taking into account the selected variables, cluster analysis was selected because it provides a set of techniques to identify subgroups

in a dataset [50]. Those techniques can be hierarchical or non-hierarchical. Hierarchical techniques are those that can agglomerate or divide elements in a set. On the other hand, the non-hierarchical technique stems from the number of clusters defined by the researcher, and finds the elements closest to them. Both cases try to calculate the minimum distance between the elements in a cluster and the maximum between clusters [51].

In this case, a well-known technique called K-means was applied to analyse those indicators for each of Poland's regions, and found subgroups depending on the level of the development of environmental and socio-economic aspects. K-means is an unsupervised machine learning non-hierarchical cluster analysis technique. From the given number of clusters, the algorithm selects the centroids and assigns each element to the closest point. Then, the coordinates of the centre are changed, and this process is repeated until there is no change in the formed clusters [52]. The K-means Python library Scikit-Learn was used to implement this kind of analysis, which a widely used tool that supports this kind of problem.

The elbow method allows selection of the ideal number of clusters for the K-means algorithm implementation. To use K-means, it was necessary as a first step to identify which was the best number of clusters in our model [53]. Then, the K-means algorithm was fitted with the data of each variable of 2011. Before that, data were scaled to values between 0 and 1, in order to reduce the outliers due to differences between measurement units of the values. After that, the algorithm was used to predict the corresponding cluster for data in 2011 and 2021, allowing us to see the changes over the period of 10 years.

To describe further the results of K-means clustering, PCA (Principal Component Analysis) was used, which is a statistical technique for reducing the dimensionality of a data set. In fact, PCA, through a linear transformation of the data into a new coordinate system, allows the description of most of the data variability by reducing the size of the initial data [54].

Furthermore, we compared the results obtained with the K-means clustering method with those obtained with the partitioning around medoids (PAM) method, in order to verify with greater reliability the migration of regions in different clusters in the considered time period. This facilitated the comparison of regions through their classification and understanding of their common characteristics. PAM is a clustering algorithm in which the K-medoids paradigm is applied [55]. Unlike K-means clustering, which uses the mean of data points within a cluster to make them the center of the cluster, PAM clustering uses data points (called medoids) that have a smaller total distance than the resulting cluster. Therefore, leaving from an initial set of medoids, if necessary, through successive iterations, it replaces them with other points if they serve to decrease the total distance of the resulting grouping.

Adopting the two clustering methods began from a key problem, which was choosing an appropriate clustering method and determining the best number of clusters. In fact, different groupings can be optimal on the same data set based on different algorithms. The comparison between the two proposed methods, K-means and PAM, better guaranteed the belonging of the regions to the clusters [56,57].

## 3. Results and Discussion

In the years 2011–2021, the value of the indicators changed (Table 1). In the case of indicator X1 (number of economic entities), growth by 24.5% on average occurred. The Mazowieckie region showed the greatest economic activity in this regard. In 2011, this indicator equaled 127.7, and in 2021, it became 171.2 entities per 1000 residents. The fewest economic entities were unchangingly present in the Podkarpackie region (71 in 2011 and 93.10 in 2021). In terms of indicator X2, two regions were distinct in 2011: Mazowieckie and Śląskie. Gross salaries in these regions were higher than the national average. In 2021, besides the Mazowieckie region (which remained the leader), a higher salaries indicator was noted in the Dolnośląskie (104) and Małopolskie (100.8) regions.

**Table 1.** Descriptive statistics, years 2011 and 2021.

| Specification | X1 | X2 | X3 | X4 | X5 | X6 | X7 | X8 | X9 |
|---|---|---|---|---|---|---|---|---|---|
| **2011** | | | | | | | | | |
| Mean | 96.48 | 92.17 | 2650.16 | 324.66 | 84.67 | 7.57 | 6.41 | 6.96 | 1.91 |
| Medium | 96.00 | 89.65 | 2593.35 | 272.72 | 76.06 | 6.41 | 6.42 | 6.50 | 1.80 |
| Min. value | 71.00 | 83.30 | 976.20 | 207.26 | 28.21 | 2.64 | 0.0 | 3.80 | 0.83 |
| Max. value | 127.70 | 124.20 | 5834.40 | 624.15 | 179.86 | 21.28 | 19.79 | 17.40 | 4.54 |
| SD | 17.45 | 10.55 | 1141.89 | 128.64 | 39.65 | 4.56 | 5.07 | 3.17 | 0.98 |
| **2021** | | | | | | | | | |
| Mean | 120.14 | 93.88 | 13,708.43 | 316.47 | 88.55 | 8.05 | 14.85 | 6.74 | 5.92 |
| Medium | 113.00 | 90.85 | 13,254.95 | 322.26 | 69.24 | 7.44 | 13.48 | 6.40 | 5.47 |
| Min. value | 93.10 | 85.40 | 10,155.90 | 209.68 | 44.74 | 3.24 | 8.07 | 3.80 | 3.08 |
| Max. value | 171.20 | 118.40 | 19,304.60 | 454.29 | 228.35 | 14.47 | 29.51 | 15.20 | 11.10 |
| SD | 22.29 | 8.61 | 2232.69 | 72.71 | 52.04 | 3.58 | 6.56 | 2.63 | 2.14 |

Source: own calculations based on [48].

The activity of regions, in terms of acquiring funds from the EU budget, increased over five-fold over the course of 10 years, and in certain regions, even ten-fold. This is due to the greater availability of funds. In recent years, substantially more programs under the Regional Operational Programme have been implemented. All regions benefited from this, but a particularly large amount of funds were directed to regions with a lower level of development as part of specially dedicated programs.

In terms of outlays for innovations (X6), changes were small during the analyzed period. The mean value of these outlays increased by 6.3%. The Mazowieckie region was distinguished in terms of outlays for innovations. In 2011, this outlay made up 21.28% of gross outlays for fixed assets. In addition to the Mazowieckie region, the Łódzkie (11.41%) and Śląskie (11.87%) regions showed outlays for innovations greater than 10% of outlays for fixed assets.

In 2021, other regions also joined the three mentioned above: the Małopolskie (14.47%), Podkarpackie (11.80%) and Pomorskie (10.72%) regions.

It was difficult to indicate a trend in the scope of environmental protection and water management (X4 and X5), as there were no unambiguous changes in indicator values during the studied period. This is probably due to the successive implementation of investments in all regions.

Indicators X7, X8 and X9 are associated with the state of transportation infrastructure. Three regions had the most roads of the highest standard (highways and express roads) in km per 100 km$^2$: Śląskie (19.79), Dolnośląskie (11.80) and Lubuskie (10.62). Two regions of Eastern Poland—Podlaskie and Podkarpackie—did not have roads of this standard in 2011, and the Lubelskie region had only 0.17 km per 100 km$^2$. In 2021, the situation changed to a more favorable one. The average status of express roads and highways in Poland increased over two-fold, and all regions since developed such roads, albeit with high differentiation.

In terms of railway lines, the Śląskie region had the best result. The length of railway lines was greatest there. The Podlaskie region had the fewest railway lines (3.8). Other regions lying on the country's eastern border also had poorly developed railway transportation (Lubelskie, Podkarpackie and Warmińsko-Mazurskie).

Indicator X9 provides information about cycle paths, and hence, indicates environmentally friendly transportation. In the years 2011–2021, the average length of cycle paths increased three-fold, and the most cycle paths were present in the Śląskie region.

It is worth noting that, in 2011, the median for all indicators was lower than the mean value, which would indicate that half of all regions exhibited a lower than average level of development with respect to the studied indicators. In 2021, the median for the majority of indicators was still lower than the mean value. Only indicators X3 and X4 did not follow this rule. This indicates that most regions increased their activity (with respect to 2011) in acquiring funds from the EU budget, and undertook investments that served

environmental protection. In the remaining cases, changes happened more or less at the same rate.

In order to carry out the cluster analysis of Poland sixteen regions based on the nine variables selected, the elbow method indicated that the ideal number of clusters was 3, according to the Figure 2. Indeed, the biggest decrease in the within-cluster sum of squares happens at 3 clusters, while the decreases are small for the models with the number of clusters greater than 3

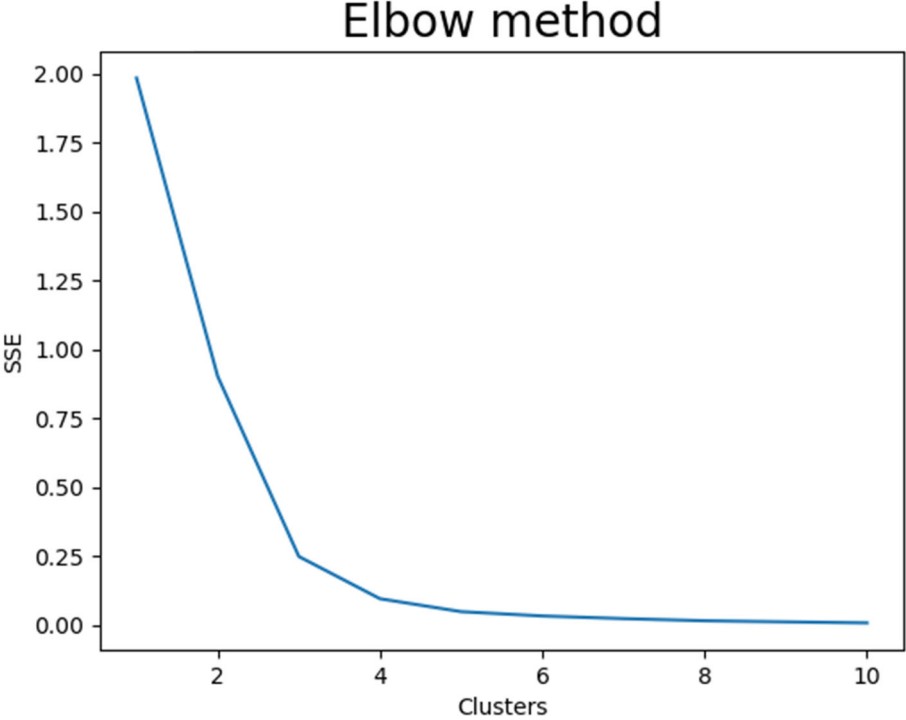

**Figure 2.** Number of correct clusters.

As we can see in Table 2, the results show the cluster assigned by the algorithm per year to each region.

There were three clusters as a result of the clustering K-means. Among the three clusters, cluster 0 included the regions that had the highest values in four out of the nine variables analyzed, and the lowest value in three out of the nine variables. Cluster 1 contained all the regions that had values in the media in four out of the nine variables, and the highest in four variables. Finally, in cluster 2 were the regions that had the lowest values in five out of the nine variables.

Figure 3 shows the classification of the regions in 2011 according to the nine indicators. The blue color represents cluster 0, where we can see highest values in indicators X1, X2, X3 and X6. Cluster 1 is represented with the red color, and the medium values in indicators X1, X2, X4 and X6, and the highest value in X5, X7, X8 and X9. Cluster 2 is represented with the green color, with the lowest values in indicators X1, X2, X6 and X9, and the highest value in X4.

There were six regions that belonged to cluster 2 in 2011. It refers to the regions that had low levels of incomes and mobility indicators, and the highest level of EU funds. With respect to other regions, those regions invested the highest amounts in environmental protection. In cluster 1, there were nine regions that had the highest level in environment protection and indicators related to mobility and sustainable means of transport. Furthermore, those regions had values in the mean with respect to the other two groups in indicators related to incomes. One region were part of cluster 0, that is, the cluster of regions with a high percentage of incomes. Moreover, this region invested a high percentage in innovation.

**Table 2.** Cluster memberships and their changes from 2011 to 2021, K-means method and PAM method.

| | | K Means | | | PAM | | | Concordance K-Means vs. PAM |
|---|---|---|---|---|---|---|---|---|
| Id | Regions | 2011 | 2021 | Change Cluster | 2011 | 2021 | Change Cluster | |
| 1 | Dolnoślaskie | 1 | 1 | no | 1 | 1 | no | ✓ |
| 2 | Kujawsko-Pomorskie | 1 | 2 | yes | 1 | 1 | no | |
| 3 | Lubelskie | 2 | 2 | no | 2 | 2 | no | ✓ |
| 4 | Lubuskie | 1 | 1 | no | 1 | 1 | no | ✓ |
| 5 | Łódzkie | 2 | 1 | yes | 1 | 1 | no | |
| 6 | Małopolskie | 1 | 1 | no | 1 | 1 | no | ✓ |
| 7 | Mazowieckie | 0 | 0 | no | 1 | 1 | no | ✓ |
| 8 | Opolskie | 1 | 2 | yes | 1 | 1 | no | |
| 9 | Podkarpackie | 2 | 2 | no | 2 | 1 | yes | |
| 10 | Podlaskie | 2 | 2 | no | 2 | 2 | no | ✓ |
| 11 | Pomorskie | 1 | 1 | no | 1 | 1 | no | ✓ |
| 12 | Śląskie | 1 | 1 | no | 3 | 3 | no | ✓ |
| 13 | Świętokrzyskie | 2 | 2 | no | 2 | 2 | no | ✓ |
| 14 | Warminsko-Mazurskie | 2 | 2 | no | 2 | 2 | no | ✓ |
| 15 | Wielkopolskie | 1 | 1 | no | 1 | 1 | no | ✓ |
| 16 | Zachodniopomorskie | 1 | 2 | yes | 1 | 2 | yes | ✓ |

Source: our processing of LDB data [48].

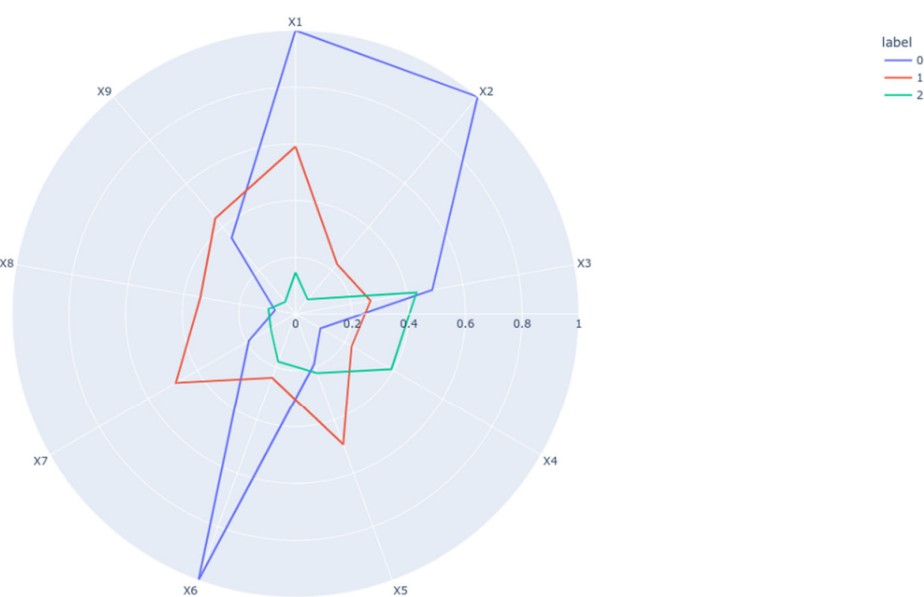

**Figure 3.** K-means clustering with all variables for 2011.

In 2021, the clusters had the same characteristics. Some regions changed their cluster; for example, one region passed to cluster 1, and three changed to cluster 2. Regarding cluster 1, one region moved up to cluster 1, which meant that it improved its indicators. Finally, cluster 2 increased in the numbers of regions because three regions decreased in the values of their indicators.

Figure 4 represents how the classification of the regions was in 2021, according to the nine indicators. Cluster 0 still had the highest values in the same indicators X1, X2 and X6, and improved the indicator X9. Cluster 1 is represented with the red color, and the medium values in X1, X2, X6 and X9 indicators meant improvement in X6, and still a higher value in X4. Cluster 2 is represented with the green color, with lowest values in indicators X1, X2, X5, X6, X7 and X9. Cluster 2 shows an improved value of X6, and still has the highest value in X3.

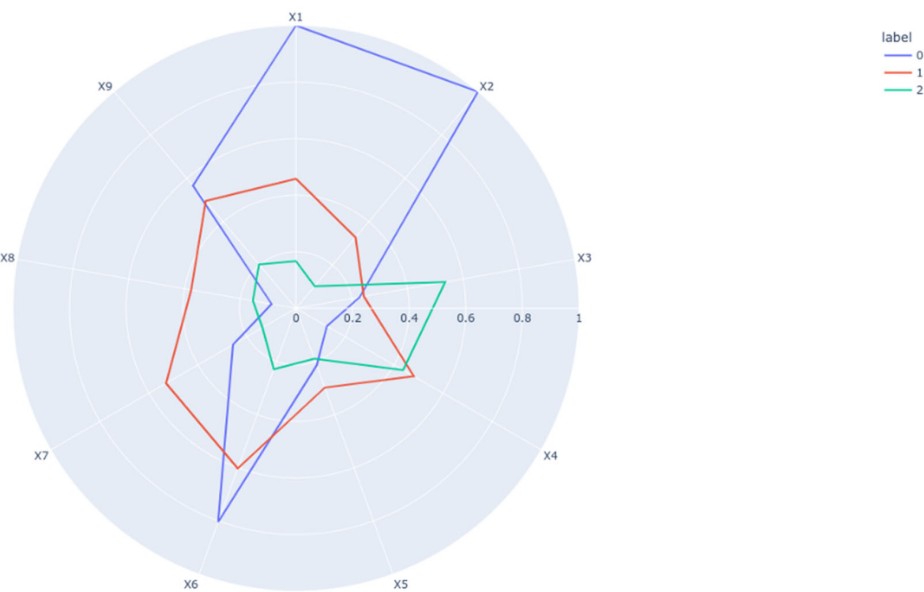

**Figure 4.** K-means clustering with all variables for 2021.

Regarding the geographical aspect, the model confirmed that there were four peripheral regions in the east part that had the lowest percentages in the majority of indicators, and the highest investments of EU funds. Regions that had the highest percentages were located near the capital city and the west side of the country.

In order to show the results of the clustering, data were represented by two types of graphics. First of all, using PCA (Principal Component Analysis), data were reduced into two variables to represent the clusters in a scatter plot in two dimensions. The results also represent the centroids of each cluster (Figures 5 and 6).

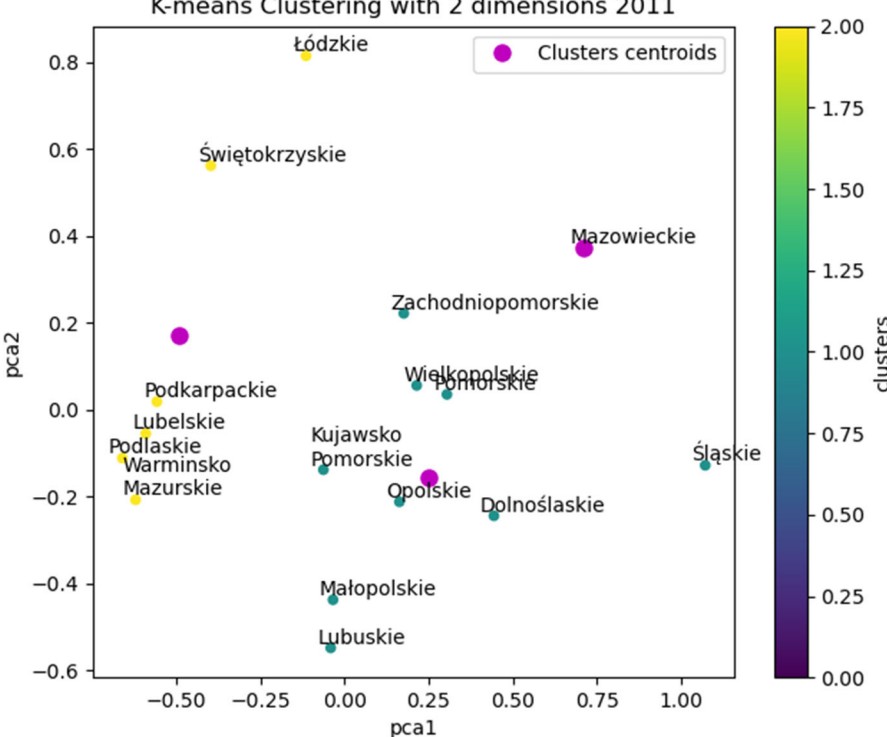

**Figure 5.** K-means clustering with 2 dimensions for 2011.

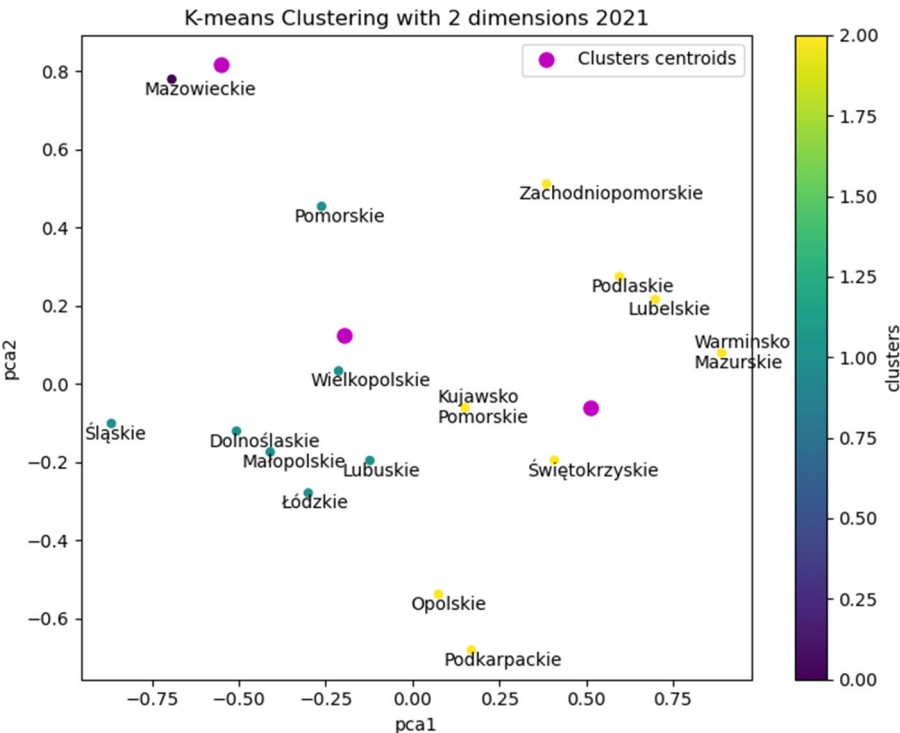

**Figure 6.** K-means clustering with 2 dimensions for 2021.

Figures 5 and 6 show the sixteen regions in their respective clusters in 2011 and 2021, respectively. Cluster 0 is represented with the blue color, cluster 1 in green, and cluster 2 in yellow.

As a result of PAM clustering, Figures 7 and 8 show the three clusters obtained for 2011 and 2021, respectively. It should be noted that there have been two migrations of regions to other clusters during the decade, affecting the Podkarpackie (id 9) and the Zachodniopomorskie regions (id 16). Regarding the comparison between the K-means and PAM methodologies in 12 cases (Table 1), there is agreement between them to confirm a region in one group or to migrate it to another.

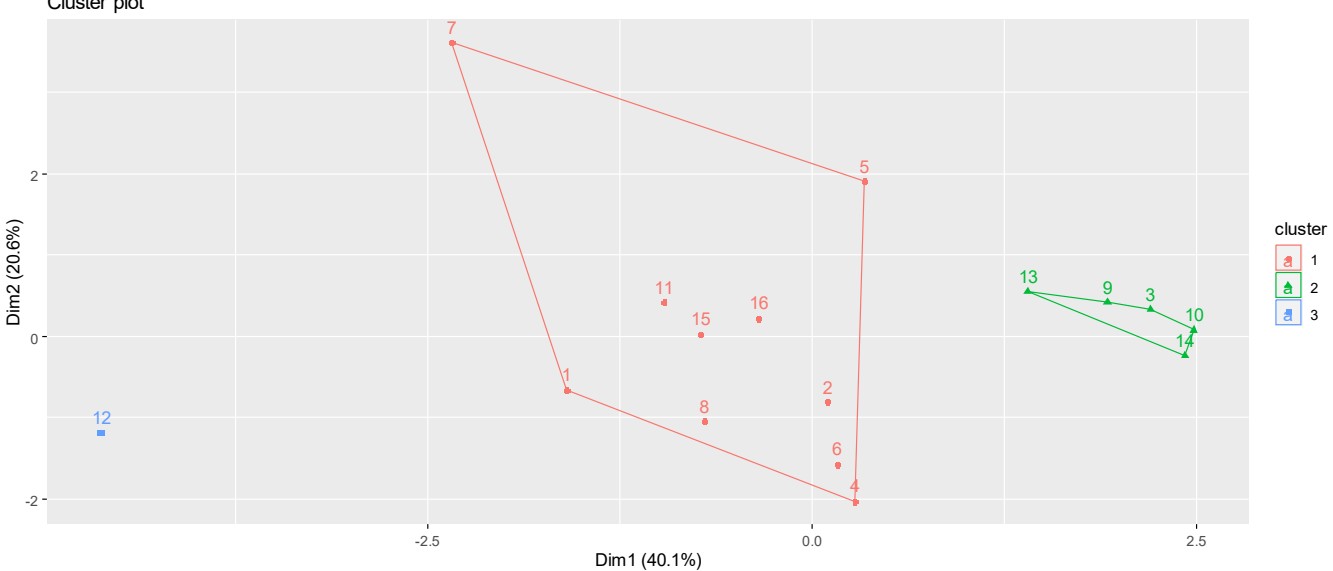

**Figure 7.** PAM clustering for 2011.

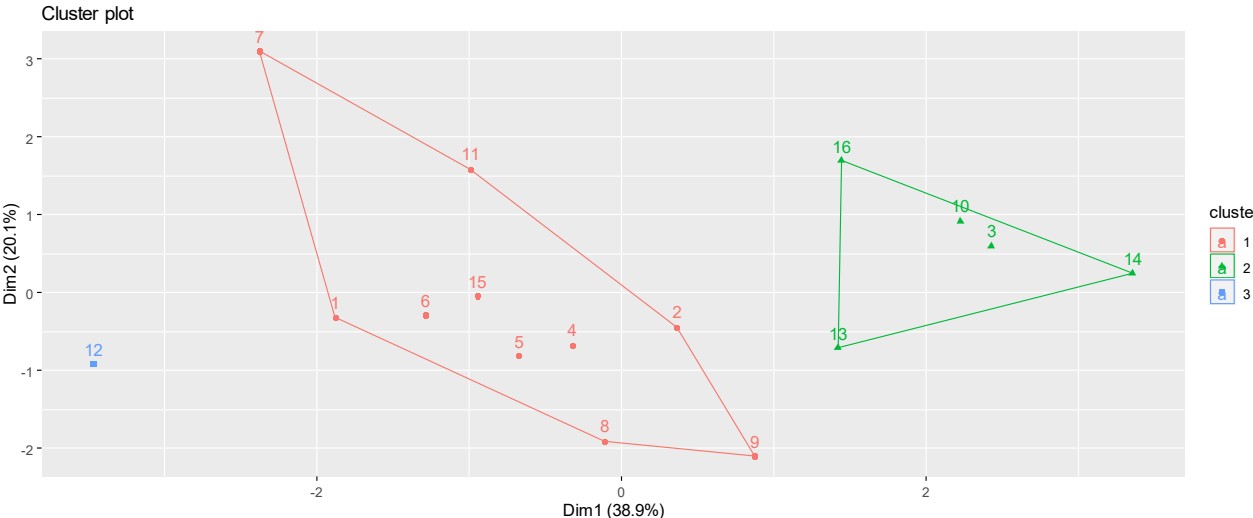

**Figure 8.** PAM clustering for 2021.

Finally, the research results show that although there has been a positive change in all regions between 2011 and 2021, this did not occur in a homogeneous way, and often the regions remained in the initial clusters without making leaps in quality to more virtuous approaches, as can be seen from Table 2 in the Change Cluster column. In fact, both cluster methodologies, with the exception of some regions, confirmed the existence of three groups characterized by indicators with internally homogeneous and heterogeneous values. The group of regions with label 2 in the K-means cluster and with label 1 in the PAM stands out: these are the most developed regions which include the Mazowieckie region, and the regions located in the western part of the country, which have maintained virtuous indicator values over some years. In a second group labeled 0 in the K-means cluster and labeled 2 in PAM, there are regions located in the eastern part of Poland, the so-called peripheral regions, which have often maintained lower standards over the years without making any leaps in quality. Finally, a third group with regions of intermediate development, indicated with 2 in the K-means cluster and with 3 in PAM like the previous group, showed no leaps in quality and migrations towards the more virtuous group.

Therefore, in line with previous research (starting from Keynes' theory), highly developed regions are becoming economically dominant, and problematic phenomena are deepening in peripheral regions. The actions of public authorities should be particularly intensive in areas of poor economic growth [28]. This is consistent with the concept of sustainable development. Such a concept has been adopted in Poland. As stated by other authors [30–32], the nature of mobility is important in the sustainable development strategy. The fact that the number of cycle paths is continuously increasing is a particularly positive sign. This indicates promotion of means of transportation that are alternative to motorized transportation. This mobility should be based on environmentally friendly means of transportation that fulfill the principles of sustainable transportation [33–35]. From a social perspective, this is very important, since improving mobility is one way of preventing marginalization and social exclusion of a significant population [13,14]. The social aspect of transportation accessibility is frequently emphasized, as it is a component of integration of local communities, and of quality of life improvement [16,18]. According to the results of a number of studies, the residents of areas with better accessibility to transport are more integrated with the local community, and are less likely to experience social exclusion [15].

## 4. Conclusions

According to the goals of sustainable development, the national development strategy should combine economic objectives with environmental and social ones. Such development should entail permanent improvements in living conditions, and competitiveness

of the regions. This is a condition for persistent and sustainable development. One of the central problems of the modern economy is the persisting differences in the level of development of individual regions.

Research results indicate that development of all regions has occurred over the years. Despite the tendency of growth, they continue to develop in a highly disproportionate manner. The division between dynamically growing regions versus peripheries is consolidating. The most developed regions are often called regions of economic success [6]. In the case of Poland, they include, above all, the Mazowieckie region (which includes the country's capital), and regions found in the western part of the country. Regions situated in the eastern part of Poland are found on the other pole. These are peripheral/marginal regions. In recent years, a particularly large amount of funds has been directed to regions with a lower level of development, as part of specially dedicated programs financed from the EU budget. At first, these programs were intended to equalize differences in socio-economic development between better-developed and peripheral regions.

Changes are visible in the improvements in mobility and accessibility; however, clear differentiation between regions persists. Peripheral regions have made great progress in this scope, as they did not have standard highways or express roads during the first period of research.

As part of the international MARA project concerning improving mobility in peripheral regions, it was stated that residents expect improvement in mobility, including public transportation, as they see in this an opportunity for broader access to the labor market and services. This is the primary component of improvements to their living conditions, as well as an opportunity to initiate social relations. Studies have also been conducted in Poland. The Strategy of Responsible Development adopted by the Polish government also places emphasis on improving the accessibility of transportation, as a factor of economic and social development. Thanks to the support of new technologies, a sustainable living space is being created in social, economic and environmental contexts [12]. Indeed, the idea of sustainable development means promoting quality of life, thus assuring people and businesses an environment full of job and firm opportunities.

In summary, public support, mainly from EU budget funds, has contributed to improvements in all indicators characterizing the level of development of Poland's regions, and these changes are really visible. The rate of these changes differs, however, and leads to consolidation of the division between rapidly developing regions and peripheral regions. This is not consistent with the principles of sustainable development. At the same time, the fact that the sustainable development concept is not conflict-free is manifesting. Economic, environmental and social sustainability criteria are difficult to fulfill simultaneously; this has been confirmed by multiple analyses. A similar situation is present in other countries, e.g., Lithuania, which is strongly polarized, with substantial regional disproportions within the country [3].

The authors are aware of the limitations arising from this analysis. These limitations comprised insufficient availability of indispensable data, and problems with determining boundary values for many indicators. Another limitation was the small number of indicators, mainly environmental and social indicators, resulting from a lack of available data.

*Future Research*

Future lines of research envisage continuing to monitor Poland's economic and environmental development over time, both by adding further indicators for more in-depth analyses, and using further clustering techniques to have greater reliability in determining whether a region belongs to a particular group.

Further studies should be expanded by other indicators, depending on the availability of data. This is important from the point of view of the effectiveness of sustainable development assessment methods in regional aspects. Indeed, a crucial aspect to be included in future research is the role of innovation for regional development. Starting from the OECD's patent statistics, the database containing patent data and indicators, and merging

the data related to social aspects obtained by national survey data, it could be possible to gain interesting insights and open new paths of research.

In contemporary urban societies, mobility and accessibility are closely linked to the fulfilment of social obligations, access to urban resources and the development and maintenance of the social networks in which the actors are inserted. Therefore, difficulties in access can translate into difficulties accessing fundamental resources for the social inclusion of individuals, such as work, welfare institutions, essential services and social ties. The hope is that the information contained in this study and in future research can provide a basis for political decision-makers, to support the weaker regions in the transformation process, in order to bring them alongside the more virtuous regions in the near future.

**Author Contributions:** Conceptualization, Z.K.-C. and M.F.; methodology, E.-L.T.-G. and A.S.; validation, Z.K.-C., A.S. and M.F.; formal analysis, E.-L.T.-G. and A.S.; investigation, E.-L.T.-G. and A.S.; resources, Z.K.-C.; data curation, A.S. and E.-L.T.-G.; writing—original draft preparation, Z.K.-C. and E.-L.T.-G.; writing—review and editing, Z.K.-C., E.-L.T.-G., M.F. and A.S.; visualization, E.-L.T.-G. and A.S.; supervision, M.F. and A.S.; funding acquisition, M.F. All authors have read and agreed to the published version of the manuscript.

**Funding:** This research received no external funding.

**Institutional Review Board Statement:** The study did not require ethical approval.

**Informed Consent Statement:** The study did not involve humans.

**Data Availability Statement:** Data supporting reported results can be found at GUS, Local Data Bank, available at https://bdl.stat.gov.pld, accessed on 19 September 2022.

**Conflicts of Interest:** The authors declare no conflict of interest.

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
