# Peer review of "Sustainable Environmental Development from the Regional Perspective—The Interesting Case of Poland"

_sustainability, doi:10.3390/su15054368_

Round 1

Reviewer 1 Report

The author used the "PAM" method to analyze how the Poland region narrowed the regional gap under the influence of the EU's regional development policy. I enjoyed the reading and review process of the entire article. In the research design, the indicators of analysis cover different professional aspects and policy-level evaluations, and boldly adopt and present unique research methods, which are worthy of praise and affirmation. The following review comments are provided for author reference:

Introduction and literature review can be merged. Please show the main purpose of your study clearly in this article.

The data collection method is worthy of admiration. However, the  method is complex, and the  analysis is too simplistic, making it easy for readers to wonder what the research focus of this article is trying to express.

The authors should provide further research questions, as well as evidence to demonstrate how "national development strategies" affect regional sustainable development effects".

The test results show the author's dedication and rigor in the research, which deserves recognition. However, it is recommended that a focused discussion of the test results be possible in the relevant discussion section before the conclusion. Otherwise, the paper will be less of a rigorous academic work and more of a book report.

The test results need to be verified with the awareness of the research problem.

"Sustainable socio-environmental development" is an important and abstract concept, and it is suggested that a clearer explanation is needed to present the data conclusions shown by statistical methods.

Proofreading is needed to correct the misspelling and other format mistakes.

It is difficult to draw the conclusions outlined in the article from the analysis design of this study, which are more like common sense knowledge.

The English writing should be improved furtherly.

If the authors can improve on the issues raised by the above and the following comments, I think this would be a worthwhile paper.

Author Response

see the file please

thanks 

Reviewer 2 Report

Dear authors,

Please refer to my comments regarding your paper.

Thanks.

Author Response

see the file please

thanks

Reviewer 3 Report

Dear authors,

Thank you for your submission. The manuscript requires a major editing as per the details in the file attached.

Best wishes!

Reviewer 4 Report

The problem of unbalanced development of regions is a significant obstacle to achieving sustainable growth of the entire national economy. This issue is particularly acute in the modern realities of environmental and technological trends. The differences in the innovation and investment attractiveness of regions depend on the specifics of each territory, which are formed under the influence of geographical, institutional, economic, technological, and other factors. Considering these factors and searching for common patterns of the socio-ecological environment of territories formation is of great scientific and practical importance for building a balanced territorial system in the country.

1. 1. In the title of the article, the key words are "socio-ecological development". The authors write that when assessing the level of regional development, indicators characterizing various aspects of Poland's development are considered: economic, social and environmental (lines 134-144). But none of the indicators they offer reflect the social direction of development. And the authors themselves, explaining the choice of indicators, refer some to economic development (lines 146-148), some to environmental development (lines 149–153), and some indicators reflect mobility and accessibility in the regions (lines 154-158), and the authors also refer to economic development. I would like to clarify which of the listed indicators the authors refer to the social aspect? If the authors did not include social factors, then how can we conclude about socio-ecological development? If the social factor is not being investigated, then maybe it's still worth correcting the title of the article?

2. Both in the introduction (lines 60-63) and in the conclusions (lines 364-367), the authors write about the role of technological innovations affecting sustainable living space in a social, economic, and environmental context, but there was no information about innovation in the study. I would like to get a justification for this. If the innovative direction is not explored in the article, then why mention the innovative aspect?

3. In the conclusions, the authors write that "Social expectations regarding mobility in Poland relate to innovative solutions, especially in relation to public transport” (lines 360-362), from which it can be concluded that the authors associate social and innovative aspects of regional development exclusively with transport. How fair is that?

4. I think that the article lacks a Future Research section that would reflect how the authors are going to further develop this topic.

In general, I evaluate both the idea of the article and the study positively, and when receiving answers to my questions, I support the publication of the article.

Round 2

Reviewer 1 Report

ok

Reviewer 2 Report

Dear authors,

Please refer to my comments as in the attachment.

Thanks.

Round 3

Reviewer 2 Report

Dears authors,

Good efforts for repairing the paper.  The quality of the paper has been improved a lot.

However, please recheck the references part as it is not well organized. Please make sure it is well organized.

All the best.